# Whole-Genome Survey Analyses Provide a New Perspective for the Evolutionary Biology of Shimofuri Goby, *Tridentiger bifasciatus*

**DOI:** 10.3390/ani12151914

**Published:** 2022-07-27

**Authors:** Xiang Zhao, Yaxian Liu, Xueqing Du, Siyu Ma, Na Song, Linlin Zhao

**Affiliations:** 1The Key Laboratory of Mariculture, Ocean University of China, Ministry of Education, Qingdao 266003, China; zx15965582296@163.com (X.Z.); duxueqing0226@163.com (X.D.); msy05011@163.com (S.M.); songna624@163.com (N.S.); 2Yantai Laishan Marine Fisheries Supervision and Monitoring Brigade, Yantai 264000, China; liuyaxn@163.com; 3Key Laboratory of Marine Eco-Environmental Science and Technology, First Institute of Oceanography, Ministry of Natural Resources, Qingdao 266061, China; 4Marine Ecology and Environmental Science Laboratory, Pilot National Laboratory for Marine Science and Technology, Qingdao 266237, China

**Keywords:** whole-genome survey, *Tridentiger bifasciatus*, molecular markers, evolutionary biology, PSMC

## Abstract

**Simple Summary:**

Gobies form a group of small fish that are extremely adaptable to their environment. However, genomic resources for Gobiidae species are scarce. In this study, high-throughput sequencing technology was used to generate the whole genome data for the shimofuri goby (*Tridentiger bifasciatus*). In addition to providing basic genomic information, the mitochondrial whole genome, microsatellite markers, and single-copy homologous genes are also explored. The PSMC model is used to provide the historical population dynamics of the shimofuri goby. The results of this study provide a reference for the assembly of a complete genome map and biological research of the shimofuri goby.

**Abstract:**

The shimofuri goby (*Tridentiger bifasciatus*) is a small and highly adaptable goby, distributed along the coasts of China, the Sea of Japan, and the west coastal and estuarine areas of the Northwest Pacific. Next-generation sequencing was used to generate genome-wide survey data to provide essential characterization of the shimofuri goby genome and for the further mining of genomic information. The genome size of the shimofuri goby was estimated to be approximately 887.60 Mb through K-mer analysis, with a heterozygosity ratio and repeat sequence ratio of 0.47% and 32.60%, respectively. The assembled genome was used to identify microsatellite motifs (Simple Sequence Repeats, SSRs), extract single-copy homologous genes and assemble the mitochondrial genome. A total of 288,730 SSRs were identified. The most frequent SSRs were dinucleotide repeats (with a frequency of 61.15%), followed by trinucleotide (29.87%), tetranucleotide (6.19%), pentanucleotide (1.13%), and hexanucleotide repeats (1.66%). The results of the phylogenetic analysis based on single-copy homologous genes showed that the shimofuri goby and *Rhinogobius similis* can be clustered into one branch. The shimofuri goby was originally thought to be the same as the chameleon goby (*Tridentiger trigonocephalus*) due to their close morphological resemblance. However, a complete mitochondrial genome was assembled and the results of the phylogenetic analysis support the inclusion of the shimofuri goby as a separate species. PSMC analysis indicated that the shimofuri goby experienced a bottleneck event during the Pleistocene Glacial Epoch, in which its population size decreased massively, and then it began to recover gradually after the Last Glacial Maximum. This study provides a reference for the further assembly of the complete genome map of the shimofuri goby, and is a valuable genomic resource for the study of its evolutionary biology.

## 1. Introduction

The shimofuri goby (*Tridentiger bifasciatus*) belongs to the Perciformes, Gobiidae and *Tridentiger*. It is a small goby distributed along the coasts of China, the Sea of Japan and the west coastal and estuarine areas of the Northwest Pacific [1]. The shimofuri goby was originally thought to be the same as the chameleon goby (*Tridentiger trigonocephalus*) due to their close morphological resemblance. In 1989, Akihito and Sakamoto identified the shimofuri goby as a separate species, based on differences in its sensory canals, pectoral fins and coloration [2]. After the 1980s, the shimofuri goby invaded the coastal waters and estuaries of the Eastern Pacific Ocean due to the development of pelagic fisheries and the carrying effect of ships’ ballast water. It was first recorded in the San Francisco Estuary in 1985, and its range rapidly expanded to become one of the most abundant fish in the region [3,4,5]. This phenomenon demonstrates the powerful adaptability of the shimofuri goby. Previous studies have shown that the shimofuri goby has a wide range of temperature and salinity tolerances, even more than the environmental adaptation ability of indigenous species in the San Francisco Estuary [6]. Its strong environmental adaptability may be an important reason as to why the shimofuri goby successfully invaded other areas. The shimofuri goby has a short lifespan, high reproductive capacity and plastic life history characteristics [7]. In addition, the dietary composition of the shimofuri goby is very complex, including mollusks, aquatic insects, other macroinvertebrates, shrimp, fish, fish eggs and detritus [8,9]. These characteristics also contribute to the ability of the shimofuri goby to survive in other areas.

To date, there have been several genetic studies of the shimofuri goby. Jin et al. assembled the whole mitochondrial genome of the shimofuri goby by primer walking and preliminarily explored the phylogenetic relationships of Gobiidae [10]. Raphalo et al. used mitochondrial cytochrome C subunit I (*COI*) and ryanodine receptor 3 (*Ryr3*) DNA sequence data to assess the genetic diversity and genetic structure of eight coastal shimofuri goby populations in Zhejiang Province, China, and showed significant genetic divergence between the southern and northern populations [11]. Qin et al. developed and characterized 14 highly polymorphic microsatellite markers in the shimofuri goby using next-generation sequencing technology; notably, these markers are important for shimofuri goby genetics research [12]. In general, shimofuri goby genetic research is still largely lacking. Although gobies have no commercial value, they are key bait organisms for commercially important fish and play an integral role in the material cycle and energy flow in ecosystems. Therefore, it is necessary to reveal its biological, genetic, and evolutionary basis.

In recent years, the rapid development of high-throughput sequencing technology and various bioinformatics analysis methods has greatly promoted the marine fish genome and evolutionary biology research [13,14]. High-throughput sequencing data with whole-genome survey analyses can provide essential information on the genome size, repeat sequence ratio, genome heterozygosity, and GC content of a study species [15,16]. Compared with traditional methods, the use of genomic data makes it more efficient to assemble organelle genomes and develop microsatellite markers [17,18]. The assembled genomic data can be used to identify single-copy homologous genes [19,20,21]; in addition, the whole genome of a single species can also be used to predict a population’s historical dynamics. In this study, the whole genome data for shimofuri goby were obtained using Illumina sequencing technology. In addition to providing basic genome data, the mitochondrial genome and single-copy homologous genes were extracted in order to reconstruct the phylogenetic relationship with other Gobiidae species. Based on the PSMC model, the historical dynamics of the shimofuri goby population were analyzed. These data greatly supplement the genetic information of shimofuri goby and provide new perspectives on the study of its genomic characteristics and evolutionary biology.

## 2. Materials and Methods

### 2.1. Sample Collection and Genome Survey Sequencing

Shimofuri goby samples were collected from Haiyang City, Shandong Province, China, in July 2021. Muscle tissues were stored in 95% ethanol at −80 °C and total genomic DNA was extracted using the standard phenol-chloroform method. The tested DNA samples were randomly broken into fragments using a Covaris ultrasonic crusher (Covaris, Woburn, MA, USA), and whole library preparation was completed by end repair, the addition of an A-tail, the addition of sequencing junction, purification and PCR amplification. The constructed libraries were sequenced using an Illumina Hiseq 2500 (San Diego, CA, USA, 150-bp paired-end reads, PE150, 350 bp insert size). The library construction and sequencing were performed at Novogene (Beijing, China). The whole-genome sequencing data were deposited in the Short Read Archive (SRA) database of National Center for Biotechnology Information (NCBI) under accession number PRJNA825009.

### 2.2. Data Analysis Methods

Quality control of raw sequencing data was performed using the FASTP software [22]. After removing low-quality reads, we randomly selected 10,000 pairs of clean reads and mapped them to the NCBI nucleotide (NT) database, displaying the top 80% of matched species. All clean reads were used to perform K-mer analysis. We used the jellyfish software to perform the K-mer calculation and generate the K-mer frequency statistics table [23]. Based on the results of the K-mer analysis, the K-mer depth distribution curve and peak were obtained and used to estimate the size of the genome. The genome size algorithm calculates: G_size_ = N_kmer_/C_kmer_, where G_size_ is the genome size, N_kmer_ is the total number of K-mer, and C_kmer_ is the peak depth. We consider the case of K-mer depth 1 as the error case, and the error rate obtained from the calculation can be used to correct the genome size. The final calculation method for the genome size is Revised G_size_ = G_size_ × (1 − Error Rate). In addition, we used the GCE software to calculate the genomic heterozygosity and proportion of repetitive sequences [24].

Filtered clean reads were assembled into the completed mitochondrial genome using the Mitofinder software [25] with a reference sequence downloaded from NCBI (https://www.ncbi.nlm.nih.gov/, accessed on 27 April 2022) under accession number NC_015992.1. Subsequently, the assembled mitochondrial sequence was annotated using the online fish mitochondrial genome database (http://mitofish.aori.u-tokyo.ac.jp/, accessed on 27 April 2022). Then, 13 protein-coding genes in mitochondrial genomes of 27 species belonging to 9 genera of Gobiidae were downloaded from the NCBI database for phylogenetic analysis (Appendix A). We concatenated the nucleotide sequences of 13 protein-coding genes. Subsequently, the tandem sequences were aligned using the muscle method in MEGA X [26], and a phylogenetic tree was constructed using the maximum likelihood method based on 1000 resamples.

We used SOAPdenovo2 to preliminarily assemble the clean reads separately into unique contigs and scaffolds [27]. In the initial assembly, we used the default parameters of the SOAPdenovo2 software (without setting the “-u” parameter) to remove alternative contigs. To further assess the assembly quality, we conducted a BUSCO analysis [28]. The Perl script “misa.pl” of the MISA software was used to identify microsatellite motifs in the de novo draft genome [29]. The search parameters were set for the detection of di-, tri-, tetra-, penta-, and hexanucleotide microsatellite motifs with a minimum of 6, 5, 5, 5, and 5 repeats, respectively. The OrthoFinder and Busco5 software were used to search for single-copy direct homologous genes [28,30], and the completed genomes of 10 species of Gobiidae (*Periophthalmus magnuspinnatus* GCA_009829125.1; *Rhinogobius similis*: GCA_019453435.1; *Mugilogobius chulae*: GCA_016735935.1; *Boleophthalmus pectinirostris*: GCA_000788275.1; *Periophthalmodon schlosseri*: GCA_000787095.1; *Scartelaos histophorus*: GCA_000787155.1; *Chaenogobius annularis*: GCA_015082035.1; *Periophthalmus modestus*: GCA_019594935.1; *Lythrypnus dalli*: GCA_011763505.1; *Lesueurigobius sanzi*: GCA_900303255.1) were downloaded from the GenBank database and used for phylogenetic analyses. Subsequently, extracted single-copy homologous genes of the shimofuri goby were annotated using the eggNOG-mapper software [31] based on eukaryotic ortholog groups (KOG) as well as the Gene Ontology (GO) and Kyoto Encyclopedia of Genes and Genomes (KEGG) databases. Analysis of population size dynamics of the shimofuri goby was carried out using the PSMC model, as implemented in the PSMC software [32]. The “fq2psmcfa” and “splitfa” tools in the PSMC software were used to create the input file for the PSMC modelling. The PSMC analysis command included the options “-N25” for the number of cycles of the algorithm, “-t15” as the upper limit for the most recent common ancestor (TMRCA), “-r5” for the initialθ/ρ, and “-p 4 + 25 * 2 + 4 + 6” for atomic intervals. The reconstructed population history was plotted using the “psmc_plot.pl” script with the substitution rate “-u 2.5 × 10^−8^” and a generation time of 1 year.

## 3. Results

### 3.1. Whole-Genome Sequencing, K-Mer Analysis and Genome Assembly

A total of 74 Gb raw data were generated by the sequencing genome survey library with 350 bp inserts. The Q20, Q30 and GC content of raw data were 96.56, 91.57 and 39.62%, respectively. A total of 73.28 Gb clean data were obtained after filtering. From the NT database mapping results, randomly selected reads were matched to the DNA of closely related species of the shimofuri goby, which proved that the data in this study did not contain obvious exogenous contamination (Figure 1). The 19-mer frequency distribution derived from the sequencing reads is plotted in Figure 2. The K-mer analyses indicated that the peak of 19-mer distribution of the shimofuri goby was at 32×, and the error rate was 0.66%; thus, the estimated genome size was 887.60 Mb. The results of the GCE software analysis showed that the heterozygosity ratio and repeat sequence ratio were 0.47% and 32.60%, respectively.

The filtered clean reads were used to assemble the draft genome. The total length, total number of sequences, max length of sequences, length of N50 and length of N90 of the contig and scaffold level genomes are shown in Table 1. The N50 length of contig and scaffold genomes were 691 and 1203, respectively. The N90 length of contig and scaffold genomes were both 127. The BUSCO analysis indicated that 25.8%, 37.5% and 36.7% of 3354 BUSCO genes were complete, fragmented and missing, respectively (see Figure 3). The average GC content of the draft genome was 38.30%. In Figure 4, the red regions represent relatively dense portions of the scatter plot.

### 3.2. Identification and Characterization of SSR for the Genome of the Shimofuri Goby

From the 818,382,753 bp scaffold-level draft genome, a total of 288,730 SSRs were identified, which included 223,927 SSR-containing sequences. However, only 44,813 sequences contained more than one SSRs, and 30,549 SSRs were present in compound formation (Table 2). The SSRs distribution frequency in this genome was estimated to be approximately 347.10 SSRs per Mb. The motif types of SSRs included 61.15% dinucleotide, 29.87% trinucleotide, 6.19% tetranucleotide, 1.13% pentanucleotide and 1.66% hexanucleotide repeats (Figure 5a). Among the dinucleotides, the most frequent motifs were AC (14.27%) and CA (13.99%), followed by AT (11.73%) and TG (11.03%) (Figure 5b). Of the trinucleotides, the most frequent motif was AAT (10.88%), while CCT (1.30%) was the least frequent trinucleotides motif (Figure 5c). Of the tetranucleotide repeats, the AAAT repeat (4.57%) was the most abundant motif, and the 5-fold repeat was predominant for all tetranucleotide repeats. (Figure 5d).

### 3.3. Identification of Single-Copy Homologous Genes in the Shimofuri Goby Genome

In this study, 856 single-copy homologous genes were identified in the shimofuri goby genome using the Busco5 software. The results of the KOG annotation demonstrated that the function of 282 single-copy homologous genes was unknown. The main known functions of single-copy homologous genes of the shimofuri goby were translation, ribosomal structure, and biogenesis, RNA processing and modification, transcription, signal transduction mechanisms, posttranslational modification, protein turnover and chaperones (Figure 6a). The results of the GO annotation showed that the main terms associated with single-copy homologous genes were cellular process, cellular anatomical entity and binding (Figure 6b). The results of KEGG annotation showed that the main pathways of single-copy homologous genes were transport and catabolism, signal transduction, translation, glycan biosynthesis and metabolism and endocrine system (Figure 6c). Both the Busco5 and OrthoFinder software were used to identify single-copy homologous genes in 10 goby species downloaded from NCBI (see Section 2 for NCBI accession numbers). We identified 295 single-copy homologous genes longer than 200 bp in the 11 goby species. OrthoFinder defaults to mafft for multiple sequence concatenation and fasttree for species evolution tree inference. The results of the phylogenetic analysis indicated that the species evolutionary tree was divided into two large branches: (1) *Periophthalmus magnuspinnatus*, *Boleophthalmus pectinirostris*, *Periophthalmodon schlosseri*, *Scartelaos histophorus* and *Periophthalmus modestus*; and (2) *Rhinogobius similis*, *Mugilogobius chulae*, *Chaenogobius annularis*, *Lythrypnus dalli*, *Lesueurigobius sanzi* and *Tridentiger bifasciatus*. *Rhinogobius similis* and *Tridentiger bifasciatus* are the closest relatives. To the best of our knowledge, this study is the first to identify single-copy homologous genes and perform phylogenetic analysis based on the whole-genome data of the shimofuri goby (Figure 7).

### 3.4. Mitochondrial Genome Assembly of the Shimofuri Goby and Phylogenetic Analysis

The complete mitochondrial genome of the shimofuri goby forms a closed circular molecule with a total length of 16,532 bp. The complete mitochondrial genome contains 13 protein-coding genes, 22 tRNA genes, 2 rRNA genes and 1 control region (or “d-loop”), as can be seen in Figure 8. Except for ND6 and eight tRNA genes (*tRNA-Gln*, *tRNA* -*Ala*, *tRNA*-*Asn*, *tRNA* -*Cys*, *tRNA*-*Tyr*, *tRNA*-*Ser*, *tRNA*-*Glu* and *tRNA*-*Pro*), which were distributed on the light strand, the rest of the mitochondrial genes were distributed on the heavy strand. Among the 13 protein-coding genes, the *COI* gene starts with GTG, ATP6 gene starts with ATA, and the other 11 genes start with ATG. Notably, 7 of the 13 protein-coding genes (*COII*, *ATP6*, *ATP8*, *ND1*, *ND3*, *ND4* and *ND6*) terminate with conventional termination codons in the mitochondrial genome, including 2 genes (*ATP6*, *ATP8*) with TAA as a termination codon, 3 genes (*ND1*, *ND3*, *ND6*) with TAG as a termination codon, and 2 genes (*COII*, *ND4*) with AGA as a termination codon. Not surprisingly, all species of *Tridentiger* were clustered together in the phylogenetic tree constructed based on 13 protein-coding genes. The phylogenetic relationships revealed the accuracy of mitochondrial genome assembly conducted in this study (Figure 9).

### 3.5. Population Size Dynamics of the Shimofuri Goby

We used the PSMC model to infer historical changes in the effective population size (Ne; see Figure 10). The PSMC results indicated that the shimofuri goby has experienced one bottleneck effect in the past million years, and its effective population size has been fluctuating. After the last interglacial period (1.1–1.3 × 10^5^ years ago; a relatively warm geological period), the effective population size reached its peak and then began to decrease continuously. Near the last glacial maximum (2.1 × 10^4^ years ago, an extremely cold geological period), the effective population size of the shimofuri goby reached its minimum, then began to recover.

## 4. Discussion

At present, whole-genome resources of Gobiidae species are relatively scarce, and whole-genome survey sequencing is expected to become a rapid and low-cost technology for evolutionary biology research. In this study, a genome-wide survey of the shimofuri goby was conducted using whole-genome sequencing. The effective rate, Q20 and Q30 of the raw data were greater than 90%, indicating that the sequencing data were of good quality and could be used for subsequent analysis. The heterozygosity ratio and repeat sequence ratio in the shimofuri goby genome were 0.47% (<0.5%) and 32.60% (<50%), respectively. Therefore, we speculate that the shimofuri goby genome belongs to the common diploid genome. The size of the shimofuri goby genome (887.60 Mb) was similar to that of the most published species of Gobiidae, being smaller than those of *Boleophthalmus pectinirostris* (955.75 Mb) and *Mugilogobius chulae* (1003.36 Mb) [33,34]. Differences in genome size may be due to the differences in repeat sequence content and transposon insertion events [35].

The obtained filtered clean reads were used to assemble the draft genome. To date, this is the first shimofuri goby genome assembled based on second-generation whole-genome sequencing data. The information of this genome provides fundamental data for research on the evolutionary biology of the shimofuri goby, as well as a reference for the further exploration of the genomic features of this species. The N50 and N90 lengths of the shimofuri goby scaffold draft genome were low relative to the complete genome of a species due to the limitations of assembling genomes using Illumina second-generation sequencing data alone [36]. It is necessary to use third-generation sequencing data and Hi-C technology in future studies in order to assemble the genome of the shimofuri goby [37,38]. The shimofuri goby scaffold draft genome was used to identify SSRs. The percentage of dinucleotide repeats was the highest and, as the repeat motif length increased, the number of loci decreased, similarly to that reported in other studies [39,40]. Previous studies have shown that long repetitive sequences have a higher mutation rate, which may lead to an increase in sequence instability, thus allowing for a decrease in the number of repeats [41]. Overall, motifs containing A or T were more abundant than those containing C or G, which is consistent with the results of genome-wide SSR studies of *Scatophagus argus* [42]. These SSR data will be important for the development of molecular markers in the shimofuri goby, although further validation studies using various populations are needed.

The shimofuri goby scaffold draft genome was used to identify single-copy homologous genes. Most of the single-copy genes belong to the housekeeper genes in the organism and, so, their functions are related to the regulation of various life activities of the organism [43]. Single-copy homologous genes are very conserved during speciation, and species formation may lead to the differentiation of homologous genes [44]. Hundreds of single-copy homologous genes obtained from genome-wide data may provide more accurate phylogenetic relationships between species than individual genetic markers [45]. The identification of single-copy homologous genes based on whole-genome survey sequencing data and phylogenetic analysis may become an effective method for the study of evolutionary relationships. Whole-genome sequencing data certainly contain not only nuclear genome sequences but also mitochondrial genome sequences. The large number of mitochondria in animal cells leads to a significantly higher mitochondrial genome sequencing depth than nuclear genes, which provided us with a sufficient amount of data to assemble a fine mitochondrial genome. The published shimofuri goby mitochondrial genome was assembled using the primer-walking method [10]. In this study, we assembled the shimofuri goby mitochondrial genome for the first time based on whole-genome sequencing data, which was consistent with the genetic composition of the published shimofuri goby mitochondrial genome. The GC content of the mitochondrial genome was 44.15%, showing obvious AT-bias, which is consistent with that observed in most fish [46]. Among the 13 protein-coding genes, the COI gene starts with GTG, the ATP6 gene starts with ATA, and the other 11 genes start with ATG. The translation initiation efficiency of these three initiation codons is high, among which ATG is the most efficient initiation codon [47]. Phylogenetic analysis based on 13 protein-coding genes showed that the shimofuri goby and the chameleon goby (*Tridentiger trigonocephalus*) were clustered together, but their evolutionary dendritic lengths did not coincide—suggesting that, although the sequences of the two species are extremely similar (0.17% sequence divergence), they did not diverge at the same time [48]. Therefore, the results of this study support the classification of the shimofuri goby as a separate species. In addition, the species of Tridentiger were closely related to the *Chaenogobius gulosus* and were divided into two large branches with species of the *Rhinogobius*. These results present some differences with species trees constructed based on single-copy homologous genes. We speculate that the 13 protein-coding genes of the mitochondrial genome have some limitations for the estimation of the evolutionary relationships between the taxonomic status of species. Furthermore, there are differences in the base substitution rates between mitochondrial genes and nuclear genes, which may cause some errors in estimating evolutionary relationships [49].

Population history dynamics have long been a hot topic of research in genetic and evolutionary biology. We used the PSMC model to infer historical changes in the effective population size of the shimofuri goby. Our results showed that the shimofuri goby experienced one bottleneck event during the Pleistocene Glacial Epoch, when its population size decreased massively and began to recover gradually after the Last Glacial Maximum. It has also been shown, in the study of Raphalo et al., that the shimofuri goby community off the coast of Zhejiang, China, may have expanded abruptly after experiencing a bottleneck event during the Pleistocene Glacial Epoch [11]; notably, the Zhejiang shimofuri goby population presented a low genetic diversity index compared to other marine fishes [11,50,51]. The accumulation of new mutations is very slow for this species, and genetic diversity is difficult to recover once it has declined [52]. Therefore, we speculate that a large amount of genetic diversity was lost in the shimofuri goby population during the contraction period. Although the population size of the shimofuri goby rapidly increased after the Last Glacial Maximum., the genetic diversity of the shimofuri goby population did not increase sufficiently, as there are highly similar “redundant copies” in the increased number of individuals from a genetic point of view.

## 5. Conclusions

In this study, the genomic survey analysis of the shimofuri goby was performed using high-throughput sequencing and genomic information, such as genome size, heterozygosity, GC content and repeat sequence ratio. We showed that the shimofuri goby genome is a common diploid genome. Furthermore, we screened a large number of SSR markers from the shimofuri goby genome. A phylogenetic tree constructed based on single-copy homologous genes clarified the phylogenetic relationships of some Gobiidae species. The assembled mitochondrial genome not only complemented the available shimofuri goby genetic resources, but also further supported its taxonomic status. PSMC analysis showed that the effective population size of the shimofuri goby decreased sharply during the Pleistocene ice age, and then recovered gradually. This study provides a reference for the further assembly of the shimofuri goby genome and offers a new perspective on its evolutionary biology. Future genome survey studies should consider six components: basic genome characterization, genome assembly, SSR marker screening, mitochondrial genome assembly, single-copy homology gene screening and PSMC analysis.

## Figures and Tables

**Figure 1 animals-12-01914-f001:**
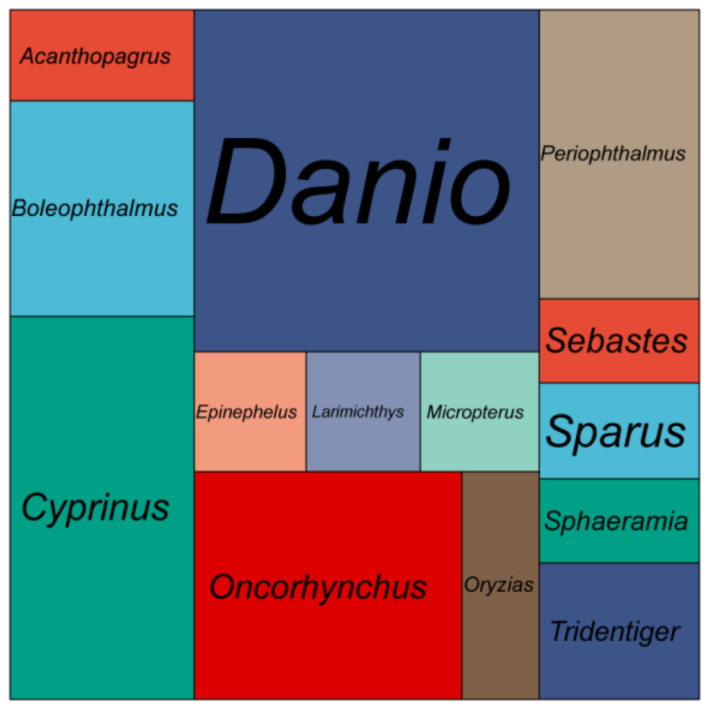
Mapping statistics of clean reads to the NT database. Area size represents the number of reads mapped to that species.

**Figure 2 animals-12-01914-f002:**
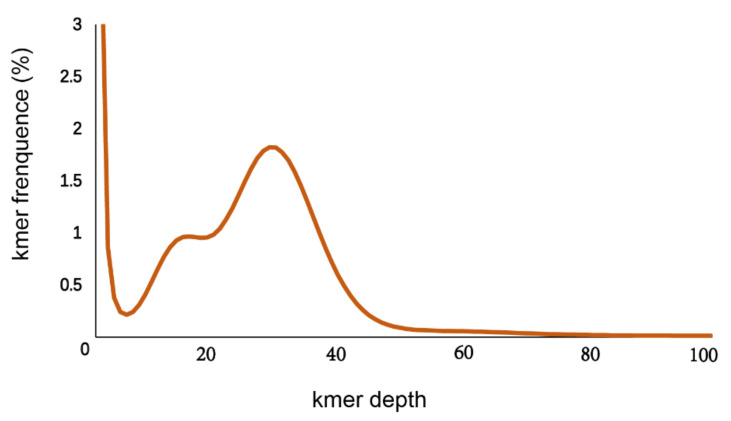
K-mer (19-mer) analysis for estimating the genome size of the shimofuri goby.

**Figure 3 animals-12-01914-f003:**
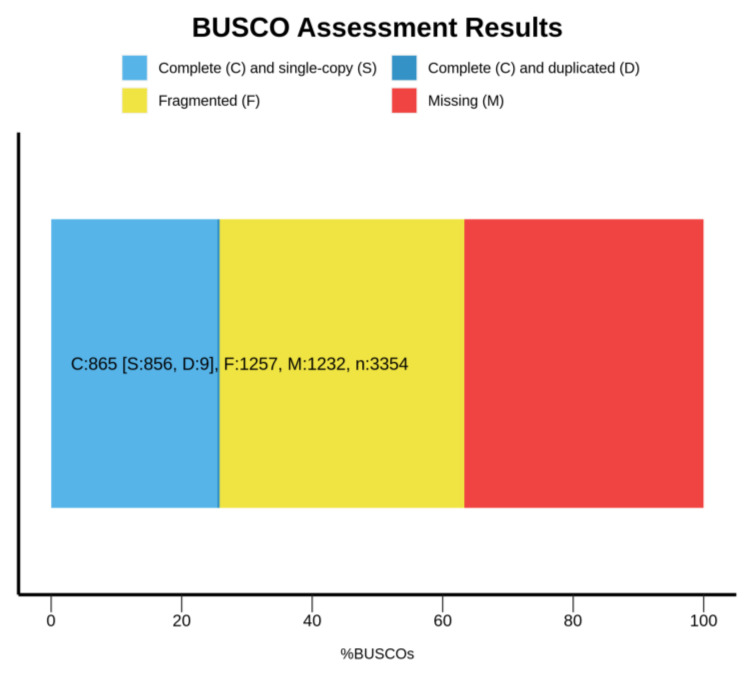
Assembly evaluation summary by BUSCO.

**Figure 4 animals-12-01914-f004:**
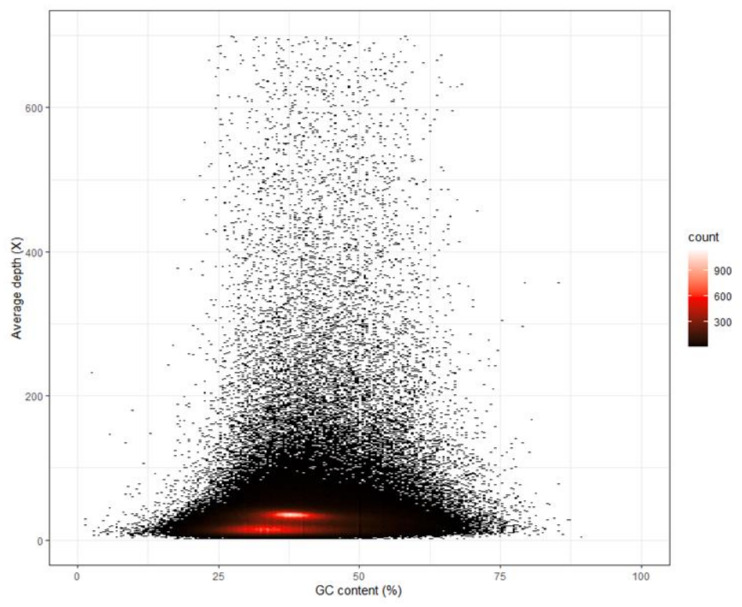
GC content and depth correlation analysis of the shimofuri goby. The *x* axis denotes the percentage GC content and the *y* axis represents sequencing depth.

**Figure 5 animals-12-01914-f005:**
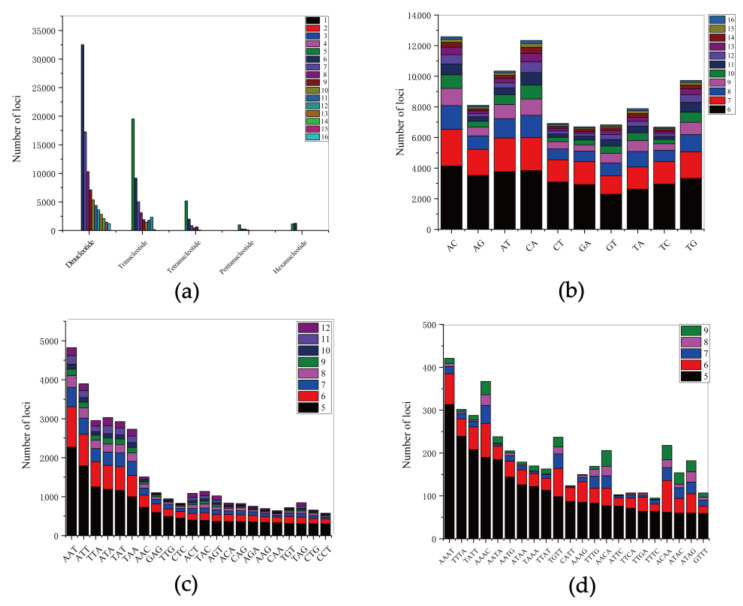
The distribution and frequency of microsatellite motifs in the shimofuri goby: (**a**) frequency of different microsatellite repeat types; (**b**) frequency of different dinucleotide microsatellite motifs; (**c**) frequency of different trinucleotide microsatellite motifs; and (**d**) frequency of different tetranucleotide microsatellite motifs.

**Figure 6 animals-12-01914-f006:**
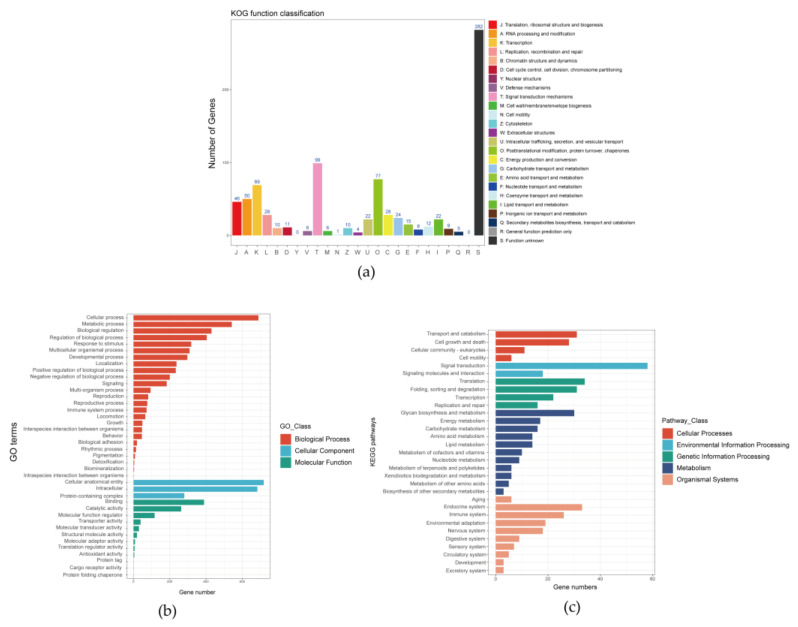
Annotation information of all single-copy homologous genes of the shimofuri goby based on three databases: (**a**) KOG classification of all single-copy homologous genes; (**b**) GO classification of all single-copy homologous genes; and (**c**) KEGG classification of all single-copy homologous genes.

**Figure 7 animals-12-01914-f007:**
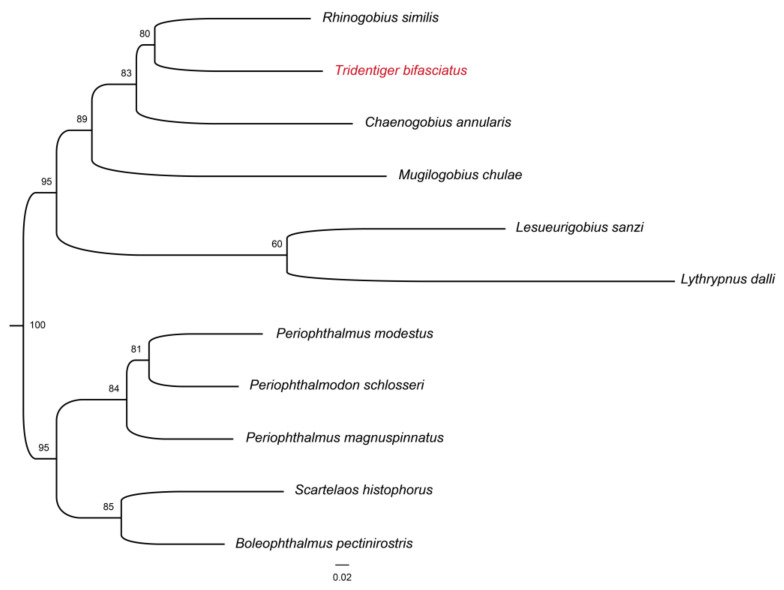
Phylogenetic tree of some species of Gobiidae based on single-copy homologous genes. Red font represents the species in this study.

**Figure 8 animals-12-01914-f008:**
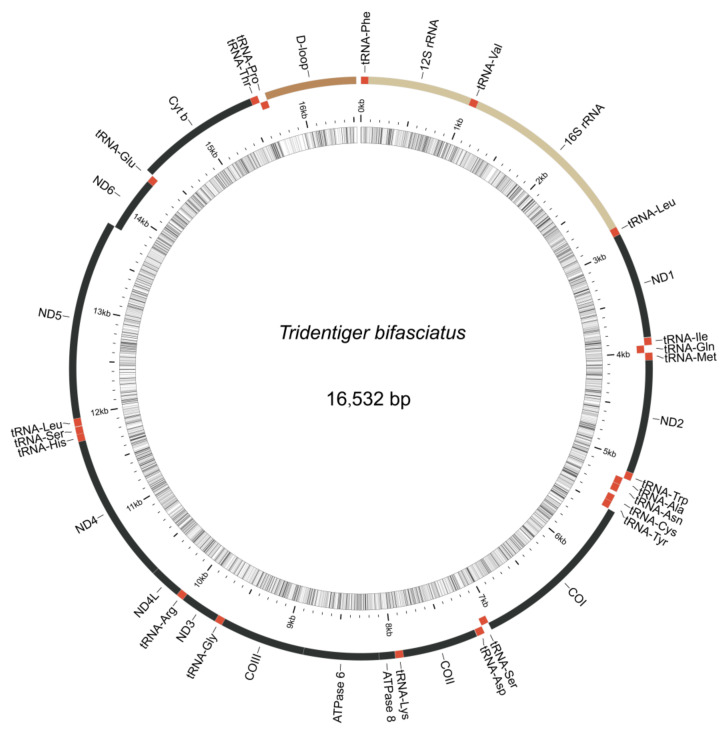
Mitochondrial genome organization of the shimofuri goby. Note: photo of the shimofuri goby from fishbase database (https://www.fishbase.de/, accessed on 27 April 2022), taken by Suzuki, Toshiyuki.

**Figure 9 animals-12-01914-f009:**
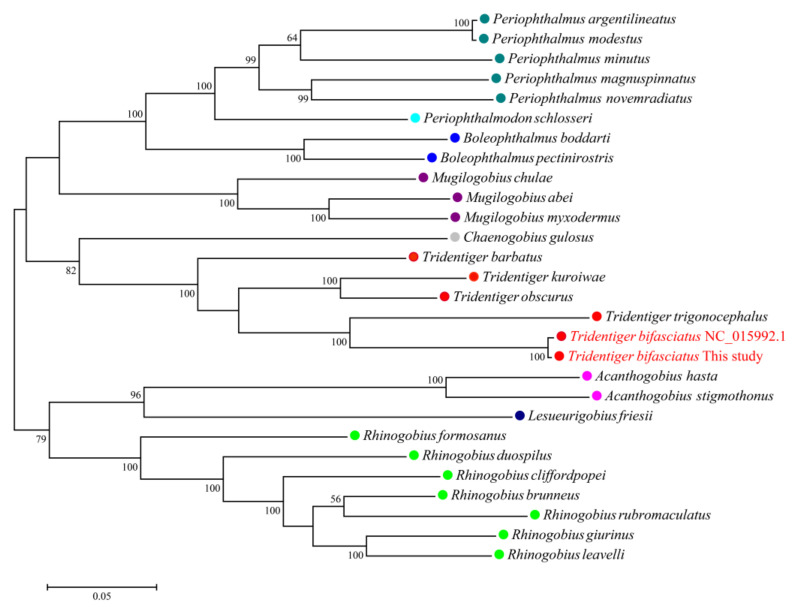
Maximum likelihood phylogenetic trees inferred from mitochondrial genomes of Gobiidae species based on 13 protein-coding genes. The same color represents species of the same genus.

**Figure 10 animals-12-01914-f010:**
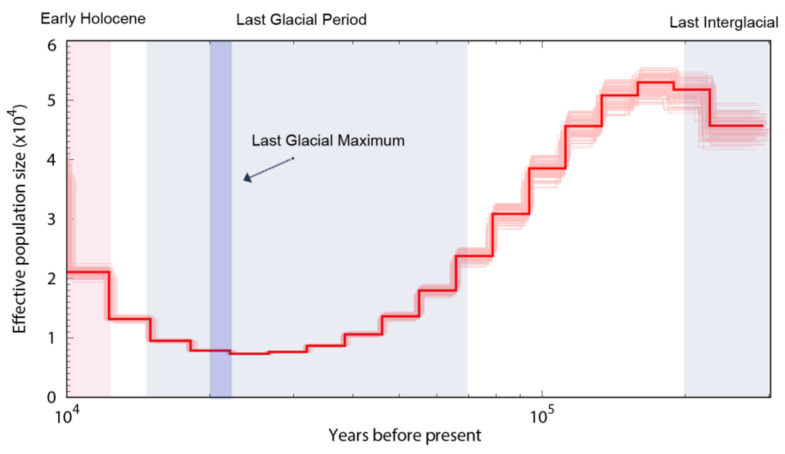
Demographic history of the shimofuri goby in this study. Note: PSMC estimates of demographic changes in effective population size (Ne) over time inferred from the draft genome sequences of the shimofuri goby. Thick lines represent the median and thin light lines correspond to 100 rounds of bootstrapping.

**Table 1 animals-12-01914-t001:** The genome result of assembly for the shimofuri goby using Illumina clean data.

	Total Length (bp)	Total Number	Max Length (bp)	N50 Length (bp)	N90 Length (bp)
Contig	830,520,379	2,382,088	25,967	691	127
Scaffold	818,382,753	1,814,709	42,478	1203	127

**Table 2 animals-12-01914-t002:** Statistics of microsatellite recognition results.

Statistical Items	Numbers
Total number of sequences examined	1,814,709
Total number of identified SSRs	288,730
Number of sequences containing more than 1 SSR	48,813
Number of SSRs present in compound formation	30,529

## Data Availability

The whole-genome sequencing data were deposited in the Short Read Archive (SRA) database of National Center for Biotechnology Information (NCBI) under accession number PRJNA825009.

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
