# Peer review of "Whole-Genome Survey Analyses Provide a New Perspective for the Evolutionary Biology of Shimofuri Goby, Tridentiger bifasciatus"

_animals, 2022, doi:10.3390/ani12151914_

Round 1

Reviewer 1 Report

Dear Authors

On the technical side, this manuscript is correct and, apart from the fact that the English wording could be improved. 

With regards to the methodology, I would ask authors to provide results for the busco and employ some tool to remove alternative contigs in their asselby (haplomerger2 or alike).

It is unclear how did authors actually sequenced the genome. They do state that Paired-End reads were employed, but they do not determine the insert size nor the read length.

The previous obserbation is very relevant, since authors later perform scaffolding with SOAPdenovo2. It's thus neccessary employing short reads (i.e. 350bp insert size) to generate contigs and larger insert sizes (i.e. mate pairs of 10Kb) to obtain scaffolds. It is unclear how they actually did this (both for the methodological and infomatic sides) and, in view of the results (low N50 and L50) these cannot be very large insert sizes.

The significance of the results presented here is in my modest opinion, very limited. Although a small niche of researchers could take advantage of these genomic results (especially of the assembly of the mitochondria), the nuclear genome statistics obtained are very poor. An N50 of 26 Kb and a L50 of 42 Kb is way below the current standars of acceptable genome assemblies nowadays. Moreover, although authors employed BUSCO for the identifiction of single-copy genes, they do not state anywhere the completeness, nor how many of these were truncated or duplicated or absent. These parameters are crucial to evaluate the quality of their assembly, and I would dare foresee some issues in this regard, such as the degree of completeness being quite poor (in view of the obtained n50). I'd ask authors to clearly give these stats in the manuscript.

I'd ask authors to undergo a major revission to clarify the busco results.

Reviewer 2 Report

Zhao et al. made a substantial effort to develop a genome assembly of an invasive goby species by using a high-throughput short read sequencing technology. This is a concise piece of a technical report, reporting basic statistics about the assembled genomes and genomic features, along with some phylogenetic and population genetic analyses. While analytical approaches meet minimal standard requirements, this may be acceptable given the nature of the study. Below are some of my comments, which, I hope, the authors find useful.

Specific comments:

Line 34: “more closely related to Rhinogobius similis” than what?

Line 34-36: Is the species status of the shimofuri goby contentious? Does this mtDNA analysis provide more definitive evidence for it to be a full species? If this is one of the aims of this study, please provide some background information in the Abstract. (I realized that this piece of information is provided in the first paragraph of the Introduction, but it is still nice to see some background information in the Abstract)

Line 106: Please add a description about the library insert size and PE read length.

Line 286-287: The present study does not provide any information about the abundance and diversity of transposable elements. I am wondering how the authors reached this inference. In addition, transposon annotation is a common exercise for this type of paper. It would be great if the authors can make some effort to annotate these important repetitive elements in the shimofuri gobi genome. If this is impractical to do for some reasons (eg., the assembled genomes are still too fragmented), please add an explanation.

Line 293-296: Can the authors provide any explanations as to why this draft genome is less contiguous than other (goby?) species despite the similar sequencing approaches?

Line 310-311: This is a misleading sentence. Divergence of homologous genes does not necessarily lead directly to species formation. A key is whether changes in a gene (or genomic sequences) lead to reproductive isolation. Divergence of homologous genes can well be a consequence of speciation (and not the direct cause of speciation).

Line 331: How extremely similar is it (in % sequence divergence)?

Line 344: I can see one bottleneck event in Figure 9 but not several.

Line 341-356: Although “the shimofuri goby (Tridentiger bifasciatus) is a small and highly adaptable goby (line 23)” distributed across a wide range of climate zones, the ancient populations of this species were heavily affected by glaciation according to this paragraph. This sounds somewhat contradictory, and I would like to hear the authors’ insights into this subject.

Line 383: Please make sure that the sequence data are available (the accession number PRJNA825009 not found in SRA as of June 28, 2022)
